# Whole Genome Resequencing Identifies Single-Nucleotide Polymorphism Markers of Growth and Reproduction Traits in Zhedong and Zi Crossbred Geese

**DOI:** 10.3390/genes14020487

**Published:** 2023-02-14

**Authors:** Guojun Liu, Zhenhua Guo, Xiuhua Zhao, Jinyan Sun, Shan Yue, Manyu Li, Zhifeng Chen, Zhigang Ma, Hui Zhao

**Affiliations:** 1Heilongjiang Academy of Agricultural Sciences, Animal Husbandry Research Institute, No. 368 Xuefu Road, Harbin 150086, China; 2Heilongjiang Academy of Agricultural Sciences, Qiqihare Branch Academy, No. 2 Heyi Road, Qiqihare 161005, China; 3Liaoning Academy of Agricultural Sciences, No. 84 Dongling Road, Shenyang 110161, China

**Keywords:** brooding, goose, whole genome resequencing, laying cycles, molecular docking

## Abstract

The broodiness traits of domestic geese are a bottleneck that prevents the rapid development of the goose industry. To reduce the broodiness of the Zhedong goose and thus improve it, this study hybridized it with the Zi goose, which has almost no broody behavior. Genome resequencing was performed for the purebred Zhedong goose, as well as the F2 and F3 hybrids. The results showed that the F1 hybrids displayed significant heterosis in growth traits, and their body weight was significantly greater than those of the other groups. The F2 hybrids showed significant heterosis in egg-laying traits, and the number of eggs laid was significantly greater than those of the other groups. A total of 7,979,421 single-nucleotide polymorphisms (SNPs) were obtained, and three SNPs were screened. Molecular docking results showed that SNP11 located in the gene *NUDT9* altered the structure and affinity of the binding pocket. The results suggested that SNP11 is an SNP related to goose broodiness. In the future, we will use the cage breeding method to sample the same half-sib families to accurately identify SNP markers of growth and reproductive traits.

## 1. Introduction

China’s domestic geese output ranks first in the world, suggesting its great potential for the development of the goose industry. However, the broodiness of domesticated geese has always been a key problem limiting the development of goose industrialization [1]. Broody behavior varies widely among different varieties of geese [2]. The broodiness of the Zhedong goose is conspicuous and seriously affects the number of eggs laid [1]. The Zi goose, which lays a large number of eggs, shows hardly any broodiness. To reduce the broodiness of the Zhedong goose, in this study we used hybridization with the Zi goose to decrease the broody behavior of the Zhedong goose. Many methods have been explored to reduce the broodiness of geese, including hormone intervention [3,4], light adjustment [5,6,7], genetic improvement [8,9], and nutrition regulation [10]. Various numbers of SNPs have been reported in species of domestic geese [11,12,13], and these can possibly be used in livestock breeding to obtain superior genotypes.

One SNP of the *growth differentiation factor 9* (*GDF9*) gene is homozygous AA in wild type ewe populations, and the mutant heterozygous Aa can increase the litter size. Interestingly, the homozygous SNP mutation aa type leads to infertility [14]. Modern breeding technology can use SNP chips to rapidly and efficiently determine the SNPs of individual animals. Animal SNP breeding chips have been developed and applied in pigs [15,16], cows [17], sheep [18], goats [19], and layers [20]. SNPs related to goose phenotypic traits have also been reported [11,12]. Currently, there are no reports concerning goose SNP chips. This study was begun in 2016 and was originally intended for commercial application. During the research, we found that the body weight and number of eggs laid by the F3 hybrids were decreased. We were concerned about *GDF9*, and thus we implemented whole genome resequencing to identify the SNP markers of growth and reproduction traits in the Zhedong and Zi crossbred geese. The purposes of this study were to verify whether there were SNPs that affected goose reproductive traits and to provide guidance for the commercial breeding of hybrid geese. These results can provide a reference for goose breeding research.

## 2. Materials and Methods

### 2.1. Ethics Statement

The study was approved by the Committee for Animal Welfare of the Institute of Animal Husbandry of Heilongjiang Academy of Agricultural Sciences, China (No. NKY-20140506), Ministry of Science and Technology.

### 2.2. Animals

The experimental geese were sourced from two farms; one was located at the Qiqihar Animal Husbandry and Veterinary Research Institute (47.35° N, 123.92° E), and the other was at the Shuangyashan Friendship Farm (46.63° N, 131.16° E). The two localities are very close; the monthly average local temperature ranges from −1 °C to 10 °C, and the minimum average temperature at night in January is −14 °C. The highest daytime average temperature in June is 37 °C. A total of 1200 Zhedong geese and 600 offspring were used as the experimental base population. In 2016 and 2019, 600 Zhedong geese were introduced from Xiangshan City, Zhejiang Province, to Heilongjiang Province. After at least one year of local domestication, they were used for hybridization experiments. The local domestication process gradually changed from the feed formula in Zhejiang Province to the feed formula used in this study. All geese were raised under natural light and temperature conditions. The geese engaged in free activities in the grounds during the day and entered their house at night. They had free access to feed and water daily. The male-to-female ratio was 1:4–5. The method of hybridization is shown in Figure 1. The nutritional standard of goose feed is listed in Table 1. Goslings hatched in May and June and started laying eggs in March of the following year.

### 2.3. Body Weight and Egg-Laying Phenotypic Measurements

The egg-laying data included a random selection of experimental mother geese from a large population. The ratio of males to females was 1:4–5, matching the male geese to form a test group. The number of eggs laid was recorded daily. The first egg was laid and recorded on 21 February 2017; thereafter February 21 each year was assigned as week 0. The bodyweight data were randomly collected from newly hatched goslings that were then marked on the neck with a dye representing a number. For example, red, yellow, blue, and green from top to bottom represented 0001. The body weight of each goose was tracked, measured, and recorded every two weeks until 12 weeks. The body weight was recorded when the goose entered and exited the goose house in order to minimize stress.

### 2.4. Genome Resequencing

Blood samples were collected for DNA extraction during the non-egg-laying stage. Five geese were randomly selected in each group to avoid as much stress as possible. Vacuum tubes containing ethylenediaminetetraacetic acid were used to collect wing blood. The blood samples were sent to Harbin Botai Gene Company for testing, and the quantity of clean data of each sample was not less than 11 GB, ensuring a Q30 value ≥ 80% (Table 2).

### 2.5. SNP Calling and GWAS

We used SnpEff to annotate the mutated sites and to determine the corresponding gene information, synonymous and non-synonymous mutations, and the impact on amino acids of the mutated sites. This study applied the classical genome-wide association study (GWAS) analysis method but failed. There were four methods: (1) ADMIXTURE (Version 1.3.0) was used to analyze the population structure; (2) The genetic relationships analysis and principal component analysis of SNPs were performed using TASSEL 5.0; (3) The PopLDdecay software(Version 3.41) was used to analyze linkage disequilibrium (LD) decay; (4) The SNP sequences were employed using Mega X to construct an evolutionary tree via the neighbor-joining (NJ) method. The 15 samples could not be clustered, and thus the analysis failed.

Finally, the filtered SNPs were further screened based on the background of the project. For the screening principle, the parents were homozygous, and the two groups of offspring samples were either the heterozygous or homozygous Zhedong goose genotype (not the homozygous Zi goose genotype). To be specific, if the primary generation was homozygous for Zhedong goose site AA, then AB and AA could appear in the F2 and F3 generations. However, BB was not possible. See Figure 1 for the annotation. These sites may be candidate SNP sites related to the phenotype of egg laying. This process used the Sort method in Microsoft Excel. GO and KEGG enrichment analyses were conducted for the SNP sites.

### 2.6. Three-Dimensional (3D) Structure Prediction and Molecular Docking

NCBI was searched for reports of candidate SNP genes in recent years. Candidate genes for *NUP37* (XP_013036270.1) and *NUDT9* (XP_013045959.1) were found. These two candidate genes include SNP3, SNP4 and SNP11. For the candidate genes, we searched for protein changes caused by SNP transformation. The 3D structures of the target proteins containing SNPs were produced via SWISS-MODEL (https://swissmodel.expasy.org). The NUDT9 protein and adenosine diphosphate ribose (ADPR) were combined. AutoDock Vina software was used for docking analysis, and the results were visualized using Discovery Studio.

## 3. Results

### 3.1. Analysis of Goose Body Weight and Egg-Laying Trait

The egg-laying data are presented in Figure 2A. From the figure, it can be seen that Zi geese began laying eggs in the middle of March and finished laying in early July. The average weekly egg-laying rate was 34.51%. From 2017 to 2020, the egg-laying rates were 17.35%, 9.07%, 14.13% and 18.65%, respectively. For the Zhedong and Zi crossed geese, the egg-laying rates of the F1, F2 and F3 hybrids were 30.05%, 70.98% and 15.02%, respectively. The duration of egg-laying for the F2 hybrids was surprising, lasting from March 4 to May 23. The birth weight (week 0) of the F3 hybrids was high, but the offspring were not used to the starter feed, resulting in a slow increase in the two-week-old body weight. When the feed was adjusted at 11 weeks, the F3 hybrids grew more rapidly. There was no significant difference between body weight and F1 hybrids at 12 weeks. The Zhedong geese in 2020 had the highest weight during the first five weeks, but this group was gradually surpassed by the F1 hybrids after changing the feed during the fifth week. At 12 weeks, the body weights of the F1 and F2 hybrids were significantly higher than those of the other experimental groups.

### 3.2. Genome Resequencing and Whole Genome Resequencing

The sequences obtained in this project were compared with the reference genome, and 7,979,421 SNPs were obtained. The SNPs obtained were filtered using VCF tools, and finally 68,376 SNPs were obtained. To further narrow the scope, it was necessary to link the SNP data with the grouped experimental data. (1) See Appendix A for the results of the population structure analysis. (2) Genetic relationship and principal component analysis results are presented in Appendix A. (3) See Appendix A for the analysis results of the linkage disequilibrium (LD) decay. The results were disappointing and could not be analyzed according to our grouping. (4) The results of the NJ evolutionary tree construction are shown in Appendix A. None of the 15 samples could be clustered. The above results show that the classical GWAS method was unable to achieve the expected result of the project.

### 3.3. Screening and Analysis of Candidate SNPs

Based on the background of this project, it was assumed that the parents were homozygous, and the two groups of offspring samples were heterozygous or homozygous for Zhedong goose sites. The offspring were grouped according to original purebred Zhedong geese, Zhedong and Zi crossbred geese, and F2, and F3 hybrid groups. A total of 3,370 candidate SNP loci were obtained. The GO enrichment analysis results are shown in Figure 3A, which lists the 25 GO terms with the highest *p*-values. These included DNA integration (GO:0015074, *p* = 4.21 × 10^−19^), the DNA metabolic process (GO: 0006259, *p* = 7.24 × 10^−12^), and the nitrogen compound metabolic process (GO: 0006807, *p* = 3.02 × 10^−6^) terms. The enrichment analysis results of level two GO terms are shown in Figure 3B. The KEGG enrichment analysis results are shown in Figure 3C. The distribution statistics of the candidate SNPs on chromosome positions are shown in Figure 3D. The 144 SNPs in the exons were further studied. Figure 3E shows the classification of the candidate SNPs distributed in the exons, and 39 SNPs caused missense mutations. These are shown in Table 3.

### 3.4. Candidate Genes’ 3D Structure Prediction and Molecular Docking

The results of SNP3 and SNP4 affecting the NUP37 protein are shown in Figure 4. The SNPs did not cause variation in protein structure, but they did change a single amino acid residue. SNP3 and SNP4 had very limited impact on the NUP37 protein. SNP11 also changed the local amino acid of NUDT9 located at the 218th amino acid of the binding site. Figure 5 shows that the amino acids were combined by 218 Thr (affinity, −5.1 kcal/mol), and the ADPR was 219 Gln, 217 Arg, and 212 Lys. The amino acids were combined by 218 Ala (affinity, −4.8 kcal/mol), and the ADPR was 210 Glu, 217 Arg, 170 Glu, and 214 Arg. SNP11 will affect the biological activity of NUDT9.

## 4. Discussion

The broodiness trait of domestic geese is a bottleneck preventing the rapid development of the goose industry [1]. The advantage of the Zhedong goose is its large body weight and superior meat quality [1]. The advantage of the Zi goose is that it has desirable egg-laying traits. Our research results show that the F1 hybrids displayed clear heterosis in growth traits, and body weight was significantly greater than those in other groups. The F2 hybrids showed significant heterosis in the egg-laying trait, and the number of eggs laid was significantly greater than those of the other groups. Our hypothesis is the SNP11 may affect the goose egg-laying trait, and the verification experiment will continue in the future.

Based on the breeding season, geese of different varieties can be divided into three types: type one and type two are long sunshine geese that inhabit the high-latitude or middle-latitude (30–40° N) temperate zone. Type three refers to the short sunshine geese distributed in subtropical regions [3]. The Zhedong goose is a short sunshine variety, and the Zi goose is a long sunshine variety. Light regulates the secretion of goose melatonin and further regulates egg-laying traits [5]. A transcriptome analysis indicated that broodiness behavior was consistent with gene expression in the pineal gland [21]. This indirectly indicated that light could change the egg-laying timing of a goose. Our results also show that the egg-laying traits of Zhedong geese changed under different light and temperature conditions when the geese migrated from Zhejiang Province to Heilongjiang Province.

The egg production of domestic geese is a quantitative trait regulated by multiple genes and is affected by environmental factors [22]. The egg-laying characteristic of the Zhedong goose comprises repeating the cycle of egg-laying and broodiness. Birds in nature also have similar behaviors [23]. Broodiness is essential for the breeding of wild birds [24], with the exception of brood parasitism by cuckoos [25]. The artificial domestication of geese is far from the effect in hens. After years of selection and elimination, hens have completely lost their broodiness ability [20].

The broodiness of the Zhedong goose is strong, and as such the goose is an ideal animal model for studying the mechanism of broody behavior [1,4,26]. In Zhejiang Province, the egg-laying period of the Zhedong goose lasts from October to April of the next year, during which three to four laying cycles (laying–broodiness–recovery) are experienced [27]. The broody behavior of a goose is regulated by the hypothalamic–pituitary–gonadal axis [28]. Many studies on goose broodiness have focused on the ovary [27,29,30,31]. The signal pathways involved in the ovarian regulation of broodiness include the autophagic pathway [29] and the GnRH pathway [32]. Our previous research showed that light affects the number of eggs laid by geese [5]. Mitochondrial dysfunction in follicles affects the broodiness of the Zhedong goose [1]. Injection of prolactin (PRL) fusion protein improved the number of eggs laid by Zhedong geese [3].

Before 2019, geese were generally raised on the ground [2]. By 2021, raising geese in cages had become popular [22]. This research started in 2016 with more than 200 geese in each group. It was impossible to select individual experimental geese based on full-sib or half-sib families. All the SNPs found included growth and reproduction traits as well as other traits. Finally, we chose the SNP screening method from the above methods. The main purpose was to gradually narrow the scope of the investigation. The first step in the screening criteria was to narrow the scope according to Mendelian genetics. The second step was to find SNPs located in the exons, and the third step was to conduct a comparative analysis according to other studies. Finally, three candidate SNPs were determined.

*NUP37* is a biomarker in breast cancer [33]. *NUP37* regulates the YAP/TEAD signaling pathway [34]. The Yes-associated protein (YAP) regulates muscle growth [35], cell division [36], and regeneration [37]. Our results showed that SNP3 and SNP4 changed an amino acid residue of NUP37, but the impact of this change needs to be further studied. This is because the change did not cause changes in the advanced structure of the NUP37 protein, and it also did not affect the structures of the binding pockets.

The newly discovered SNP11 in this study is located in *NUDT9*, a gene that can regulate the menstrual cycle [38]. This suggests that SNP11 is an SNP related to goose broodiness. *NUDT9* plays a role in breast cancer [39]. The NUDT9 protein can dock with ADPR, and 218 amino acids directly participate in the docking [40]. We simulated the docking and found that the goose NUDT9 protein could dock with ADPR. Neither 218 Thr nor 218 Ala corresponding to SNP11 directly participated in the link, but the structures of the binding pockets were altered. The 217 and 219 amino acids participated in the docking, and the affinity was altered.

## 5. Conclusions

In terms of growth traits, the F1 hybrids of Zhedong and Zi crossbred geese showed significant heterosis, and the body weight was significantly greater than in other groups. The F2 hybrids showed significant heterosis in egg-laying traits, and the number of eggs laid was significantly greater than those of the other groups. SNP11, located in *NUDT9*, is an SNP related to goose breeding. Moving from Zhejiang Province to Heilongjiang Province caused the egg-laying performance of the Zhedong goose to change under different lighting and temperature conditions. In the future, we will use the cage breeding method to sample the same half-sib family to accurately identify the SNP markers of growth and reproduction traits.

## Figures and Tables

**Figure 1 genes-14-00487-f001:**
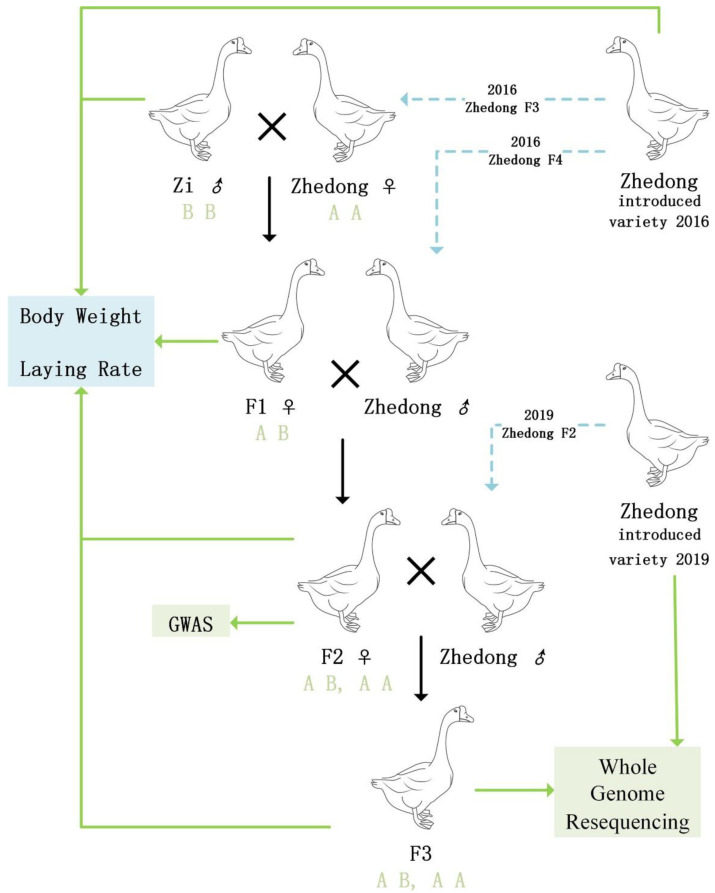
Schematic diagram of Zhedong and Zi crossbred goose experiment. Purebred Zhedong geese were introduced in Xiangshan County in 2016 and 2019. The Zi goose is a local variety in Heilongjiang. The F1, F2 and F3 hybrids were obtained through hybridization, and some experimental populations were selected to measure body weight and laying rate as well as for whole genome resequencing analysis. If the primary generation Zhedong goose was homozygous for site AA, then gene type AB and AA could appear in the F2 and F3 generations. However, BB type was not possible.

**Figure 2 genes-14-00487-f002:**
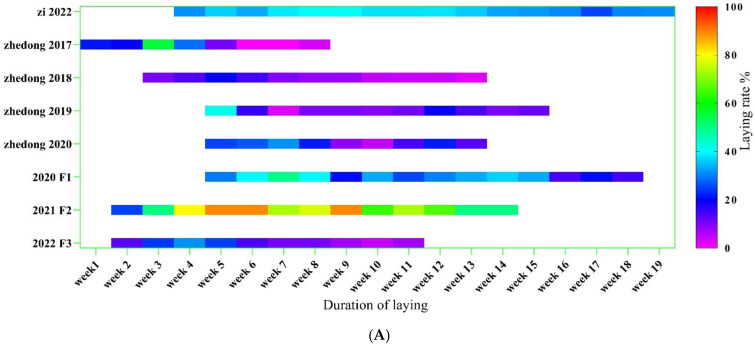
Changes in the goose egg-laying trait and body weight in different generations. (**A**) Heat map of goose egg-laying data of different generations. The different colors represent different laying rates, and the abscissa is initially dated 21 February. The end time is 3 July; the total number of days is 133. On 29 February 2020, the experimental group did not start laying eggs. (**B**) Growth data of different generations of geese (0–4 weeks) (**C**) Growth data of different generations of geese (6–8 weeks) (**D**) Growth data of different generations of geese (10–12 weeks). The body weight of the F3 hybrids shows significant differences at 2 weeks and 12 weeks. The different letters indicate significant differences. The green bar represents SE. Body-weight unit: g.

**Figure 3 genes-14-00487-f003:**
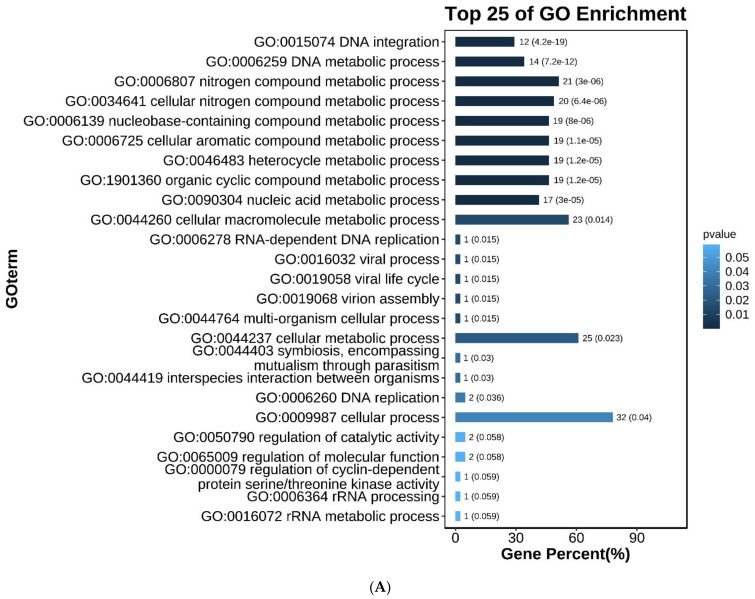
Screening and analysis of candidate SNPs. (**A**) Gene Ontology (GO) analysis for 3.370 candidate SNPs. (**B**) Level two GO analysis of candidate SNPs. (**C**) KEGG enrichment of candidate SNPs. (**D**) Distribution statistics of candidate SNPs on chromosome positions. (**E**) Classified statistics of candidate SNPs distributed in the exon.

**Figure 4 genes-14-00487-f004:**
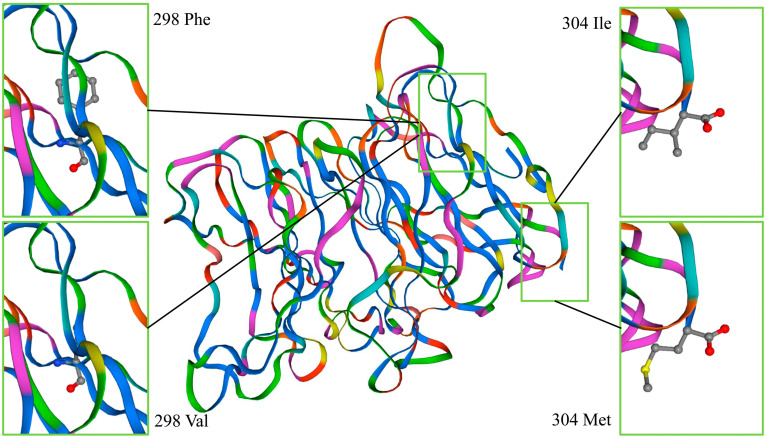
SNP3 and SNP4 affect the NUP37 protein. The structure of the NUP37 protein was unchanged due to SNP3 and SNP4. The 304th amino acid of SNP3 in the upper right corner is Ile, and in the lower right corner is Met. The 298th amino acid of SNP4 in the upper left corner is Phe, and in the lower left corner is Val.

**Figure 5 genes-14-00487-f005:**
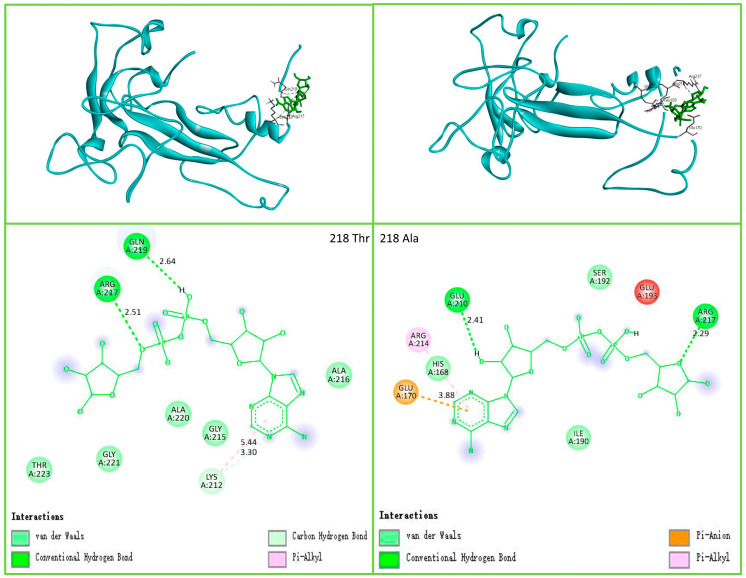
SNP11 affects NUDT9 molecular docking. SNP11 does not affect the structure of the NUDT9 protein. To display the docking of amino acids, different angles were rotated. The left side shows 218 Thr combined with ADPR. The right side shows 218 Ala combined with ADPR. The combination box has changed.

**Table 1 genes-14-00487-t001:** Goose Feed Nutrition Standard.

Content	0–4 Week	5–10 Week	11–28 Week	Adult Goose (Winter)	Egg-Laying Goose
Energy MJ/kg	11.7	11.7	11.1	10.7	11.3
Crude protein %	18	15.5	12	8	15.5
Ca%	1	1.2	1.2	1.2	2.35
Lysine %	1	1	0.7	0.7	0.75

Methionine 0.35%, phosphorus 0.65%, salt 0.35%, iron 25mg/kg, zinc 90 mg/kg, selenium 90 mg/kg. The allowable error of the above nutrients is ±10%.

**Table 2 genes-14-00487-t002:** Sequencing data statistics.

Sample	Total Reads	Unaligned Reads	Uniquely Aligned Reads	Clean_Paired_Reads	GC (%)	Q30 (%)
B311	73,532,158	7,269,362 (9.88596%)	63,142,228 (85.8702%)	36,787,767	42%	95.67%
B314	72,163,278	6,729,461 (9.32533%)	62,789,055 (87.0097%)	36,081,639	42%	95.68%
B316	73,688,664	6,925,219 (9.39794%)	63,955,897 (86.792%)	36,844,332	42%	95.66%
B319	73,511,268	6,445,136 (8.76755%)	64,116,644 (87.2202%)	36,755,634	42%	96.24%
B320	73,829,012	6,502,035 (8.80688%)	64,631,140 (87.5417%)	36,914,506	42%	95.57%
C401	73,517,946	6,853,528 (9.32225%)	63,752,893 (86.7175%)	36,758,973	42%	96.40%
C402	73,428,654	7,519,679 (10.2408%)	62,603,346 (85.2574%)	36,714,327	42%	96.79%
C407	73,539,408	6,771,803 (9.2084%)	63,947,268 (86.9565%)	36,769,704	42%	96.35%
C411	65,064,624	6,636,293 (10.1995%)	55,704,018 (85.6134%)	32,532,312	42%	91.20%
C412	73,430,222	6,913,259 (9.41473%)	63,126,236 (85.9676%)	36,715,111	42%	96.10%
O1102	73,749,910	6,707,802 (9.09534%)	64,160,474 (86.9974%)	36,874,955	42%	95.32%
O1112	73,671,940	6,572,557 (8.92138%)	64,163,844 (87.094%)	36,835,970	42%	95.45%
O1114	73,888,590	6,678,570 (9.0387%)	64,367,737 (87.1146%)	36,944,295	42%	95.67%
O1116	73,820,700	6,682,537 (9.05239%)	64,336,901 (87.1529%)	36,910,350	42%	95.52%
O1125	73,684,588	6,743,275 (9.15154%)	64,208,342 (87.1394%)	36,842,294	42%	95.68%

C represents the original Zhedong goose, B represents F2, and O represents F3.

**Table 3 genes-14-00487-t003:** A total of 39 candidate SNPs were screened in this study.

	CHROM	POS	Gene_ID	Transcript_ID	Codan_Mutate	AA_Mutate
SNP 1	NW_013185657.1	483,398	106040530	XM_013188477.1	c.479G > A	Arg160Gln
SNP 2	NW_013185677.1	5,511,334	106033575	XM_013177200.1	c.3523C > A	Gln1175Lys
SNP 3	NW_013185696.1	595	106035773	XM_013180816.1	c.912G > A	Met304Ile
SNP 4	NW_013185696.1	615	106035773	XM_013180816.1	c.892T > G	Phe298Val
SNP 5	NW_013185720.1	3,081,938	106038401	XM_013184886.1	c.977G > A	Cys326Tyr
SNP 6	NW_013185720.1	3,082,790	106038401	XM_013184889.1	c.245G > A	Arg82Gln
SNP 7	NW_013185721.1	2,609,554	106038498	XM_013185039.1	c.1637A > G	Gln546Arg
SNP 8	NW_013185748.1	3,578,791	106040672	XM_013188695.1	c.522C > G	Phe174Leu
SNP 9	NW_013185750.1	69,680	106040872	XM_013189032.1	c.1442T > G	Ile481Ser
SNP 10	NW_013185760.1	14,289	106041536	XM_013190069.1	c.3349G > T	Val1117Phe
SNP 11	NW_013185766.1	956,541	106041864	XM_013190505.1	c.652A > G	Thr218Ala
SNP 12	NW_013185766.1	956,750	106041864	XM_013190505.1	c.861A > G	Ter287Ter
SNP 13	NW_013185800.1	639,495	106043465	XM_013192926.1	c.290G > A	Arg97Lys
SNP 14	NW_013185811.1	2,202,032	106043881	XM_013193607.1	c.509G > C	Arg170Thr
SNP 15	NW_013185811.1	2,202,034	106043881	XM_013193607.1	c.507G > C	Glu169Asp
SNP 16	NW_013185864.1	1,205,891	106045958	XM_013196755.1	c.298C > T	Leu100Phe
SNP 17	NW_013185870.1	1,328,057	106046235	XM_013197144.1	c.353C > T	Ala118Val
SNP 18	NW_013185914.1	62,943	106047476	XM_013198970.1	c.561A > C	Arg187Ser
SNP 19	NW_013185914.1	63,115	106047476	XM_013198970.1	c.389A > G	Glu130Gly
SNP 20	NW_013185915.1	942,834	106047517	XM_013199056.1	c.163A > G	Met55Val
SNP 21	NW_013185915.1	942,936	106047517	XM_013199056.1	c.265A > G	Thr89Ala
SNP 22	NW_013185925.1	822,558	106047720	XM_013199342.1	c.512G > A	Arg171His
SNP 23	NW_013186010.1	166,638	106049132	XM_013201311.1	c.92G > C	Gly31Ala
SNP 24	NW_013186010.1	166,674	106049132	XM_013201311.1	c.56G > A	Arg19His
SNP 25	NW_013186010.1	261,219	106049145	XM_013201321.1	c.25G > A	Ala9Thr
SNP 26	NW_013186017.1	261,566	106049239	XM_013201442.1	c.95A > C	Tyr32Ser
SNP 27	NW_013186061.1	144,964	106049600	XM_013201966.1	c.134A > C	Asn45Thr
SNP 28	NW_013186074.1	48,559	106049689	XM_013202086.1	c.620G > A	Arg207His
SNP 29	NW_013186089.1	144,070	106049779	XM_013202193.1	c.13A > T	Thr5Ser
SNP 30	NW_013186111.1	71,554	106049894	XM_013202325.1	c.422C > A	Thr141Asn
SNP 31	NW_013186111.1	71,555	106049894	XM_013202325.1	c.421A > C	Thr141Pro
SNP 32	NW_013186150.1	35,471	106029399	XM_013170682.1	c.397G > A	Ala133Thr
SNP 33	NW_013186231.1	6,643	106029521	XM_013170813.1	c.349G>T	Ala117Ser
SNP 34	NW_013186264.1	9,000	106029548	XM_013170832.1	c.303G > C	Leu101Phe
SNP 35	NW_013186857.1	861	106029642	XM_013170927.1	c.680T > C	Ile227Thr
SNP 36	NW_013187000.1	470	106029654	XM_013170934.1	c.200T > C	Ile67Thr
SNP 37	NW_013187000.1	667	106029654	XM_013170934.1	c.397A > G	Ile133Val
SNP 38	NW_013187000.1	1,090	106029654	XM_013170934.1	c.820G > A	Val274Ile
SNP 39	NW_013188772.1	217	106029704	XM_013170974.1	c.341G > A	Gly114Asp

## Data Availability

Not applicable.

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
