# Peer review of "Whole Genome Resequencing Identifies Single-Nucleotide Polymorphism Markers of Growth and Reproduction Traits in Zhedong and Zi Crossbred Geese"

_genes, 2023, doi:10.3390/genes14020487_

Round 1

Reviewer 1 Report

The manuscript “Genome-wide association study identifies single-nucleotide polymorphism markers of growth and reproduction traits in Zhedong and Zi crossbred goose” is interesting as it does genome resequencing in purebred Zhedong goose showing that F1 hybrids had heterosis in growth traits, having better body weight, and F2 hybrids laid more eggs. These findings are very interesting for geese production, but the presentation of the manuscript needs to be greatly improved. Before being able to correct specific mistakes, the authors must restructure the manuscript according to some general observations. 

First of all, the introduction needs to be less repetitive (the word broodiness is repeated 6 times in 8 lines) and adequately justify the research, without the results, the aim should be clear. Amply explain why the research is necessary and how it could help the issues found with geese broodiness.

The materials and methods should explain all the performed procedures and used materials, and remove the purpose of the experiment explained in the Genome Sequencing paragraph. Also, the statistical analysis should be clearly stated. 

The results should only mention what was found, some parts would be better placed in materials and methods, for example, the grouping of the original purebred Zhedong geese.

The discussion is inadequate, it is a mixture of introduction, as it has elements that justify the research, and results, but it doesn’t discuss the findings and the reasons for them amply enough, it needs to be restructured. 

Lines should be numbered to be able to correctly reference which part of the manuscript needs improvements. 

Author Response

Revised manuscript 2140918 " Genome-wide association study identifies single-nucleotide polymorphism markers of growth and reproduction traits in Zhedong and Zi crossbred goose"

Dear Editor-in-Chief Pro Ziana Zhang, and Reviewers:

On behalf of my co-authors, we thank you very much for giving us an opportunity to revise our manuscript, we appreciate editor and reviewers very much for their positive and constructive comments and suggestions on our manuscript.

The manuscript has already been corrected and is sent to you for further evaluation.

We would like to express our great appreciation to you and reviewers for comments on our paper. Looking forward to hearing from you.

It's a very worrying time for all of us. I hope you are all in good health, and hoping the Coronavirus pandemic will soon be over.

Thank you and best regards.

Yours sincerely,

Guo

Animal Husbandry Research Institute of Heilongjiang Academy of Agricultural Sciences

368 Xuefu Road, Harbin, P.R.China, 150086

Office Tel: 086-451-87502330

Mobile:  086-13115607125

Responses to Reviewers

Reviewer 1

The manuscript “Genome-wide association study identifies single-nucleotide polymorphism markers of growth and reproduction traits in Zhedong and Zi crossbred goose” is interesting as it does genome resequencing in purebred Zhedong goose showing that F1 hybrids had heterosis in growth traits, having better body weight, and F2 hybrids laid more eggs. These findings are very interesting for geese production, but the presentation of the manuscript needs to be greatly improved. Before being able to correct specific mistakes, the authors must restructure the manuscript according to some general observations.

  1. First of all, the introduction needs to be less repetitive (the word broodiness is repeated 6 times in 8 lines) and adequately justify the research, without the results, the aim should be clear. Amply explain why the research is necessary and how it could help the issues found with geese broodiness.

RESPONSE:

Line 35, the statements of “The broodiness of the” was changed by “Broody behavior varies widely among”.

Line 42, the statements of “broodiness trait” was changed by “broody behavior”.

Line 18 the statements of “broodiness” was changed by “broody behavior”.

Line 223, the statements of “broodiness” was changed by “broody behavior”.

Line 51, add the statements of “The purposepurposes of this study iswere to verify whether there arewere SNPs that af-fectaffected goose reproductionreproductive traits, and to provide guidance for us to commercializethe commercial breeding of hybrid geese.”

  1. The materials and methods should explain all the performed procedures and used materials, and remove the purpose of the experiment explained in the Genome Sequencing paragraph. Also, the statistical analysis should be clearly stated.

RESPONSE:

Line 78, add the statements of “The local domestication process gradually changed from the feed formula in Zhejiang Province to the feed formula used in this study.”

Line 99, delete the statements of “The initial purpose of the experiment was to improve the number of eggs laid by the Zhedong goose. The number of eggs laid by the F2 hybrids was surprising. When we found that the number of eggs laid by the F3 hybrids has decreased, we were worried that a homozygous SNP can improve the egg-laying trait, but a heterozygous SNP can reduce the egg-laying trait. Therefore, genome resequencing was performed for the purebred Zhedong goose and F2 and F3 hybrids.”

Line 108, add the statements of “This study applied the classical GWAS analysis method but failed.”

Line 123, add the statements of “This process usesused the Sort method ofin Microsoft Excel.”

  1. The results should only mention what was found, some parts would be better placed in materials and methods, for example, the grouping of the original purebred Zhedong geese.

RESPONSE:

Thank you for your suggestion.

The grouping of the original purebred Zhedong geese in Heilongjiang, that is the first data measured by our team. So we still want to keep it in the result section.

  1. The discussion is inadequate, it is a mixture of introduction, as it has elements that justify the research, and results, but it doesn’t discuss the findings and the reasons for them amply enough, it needs to be restructured.

RESPONSE:

Line 42, add the statements of “DifferentVarious numbers of SNPs have been reported in species of domestic geese have different SNPs reported [3, 4, 5]. The purpose of , and these can possibly be used in live-stock breeding is to obtain better genotypesuperior genotypes.”

Line 297, add the statements of “5.      Gao G, Chen P, Zhou C, Zhao X, Zhang K, Wu R, Zhang C, Wang Y, Xie Y, Wang Q. Genome-wide association study for reproduction-related traits in Chinese domestic goose. Br Poult Sci 2022 63, 754-760.”

Line 214, add the statements of “The egg production of domestic geese is a quantitative trait regulated by multiple genes, which and is affected by environmental factors [22]. The egg-laying characteristic of zhedongthe Zhedong goose is to repeatcomprises repeating the cycle of egg-laying and broodiness. BirdBirds in nature also hashave similar behaviorbehaviors [23]. Broodiness is essential for the breeding of wild birds [24], exceptwith the exception of brood parasit-ism by cuckoos [25]. The artificial domestication of geese is far from the effect ofin hens. The artificial domestication of geese is far from the effect of hens. After years of selection and elimination, hens have completely lost their broodiness ability[12].”

Line 357, add the statements of “22.    Gao G, Gao D, Zhao X, Xu S, Zhang K, Wu R, Yin C, Li J, Xie Y, Hu S, Wang Q. Genome-Wide Association Study-Based Identification of SNPs and Haplotypes Associated With Goose Reproductive Performance and Egg Quality. Front Genet 2021 12, 602583.”

Line 361, add the statements of “23.    Farrar VS, Flores L, Viernes RC, Ornelas Pereira L, Mushtari S, Calisi RM. Prolactin promotes parental responses and alters reproductive axis gene expression, but not courtship behaviors, in both sexes of a biparental bird. Horm Behav 2022 144, 105217.”

Line 365, add the statements of “24.    Norris AR, Martin K, Cockle KL. Weather and nest cavity characteristics influence fecundity in mountain chickadees. PeerJ 2022 10, e14327.”

Line 368, add the statements of “25.    Morelli F, Benedetti Y, Pape Moller A. Diet specialization and brood parasitism in cuckoo species. Ecol Evol 2020 10, 5097-5105.”

  1. Lines should be numbered to be able to correctly reference which part of the manuscript needs improvements.

RESPONSE:

This is a very important idea. Thank you very much.

Added line number.

Reviewer 2

Although the work appears novel there are too many concerns that need to be addressed before the paper can be accepted.

  1. Firstly the title says GWAS but no details of GWAS or the results are presented. GWAS was performed by which software, what model was used?

RESPONSE:

Line 108, add the statements of “This study applied the classical GWAS analysis method but failed.”

We try use Mega X, TASSEL and ADMIXTURE, but all result can not use.

Line 116, the statements of “Finally, the filtered SNPs were further screened based on the background of the project. Screening principle: the parents are homozygous, and the two groups of offspring samples are heterozygous or homozygous Zhedong goose (not homozygous Zi goose genotype). To be specific, if the primary generation is homozygous to Zhedong goose site AA, then AB and AA can appear in the F2 and F3 generations. However, BB is not possible. See Figure 1 for the green annotation. These sites may be candidate SNP sites related to the phenotype of egg-laying.” Was changed by “Finally, the filtered SNPs were further screened based on the background of the pro-ject. ScreeningFor the screening principle:, the parents arewere homozygous, and the two groups of offspring samples are were either the heterozygous or homozygous Zhedong goose genotype (not the homozygous Zi goose genotype). To be specific, if the primary generation iswas homozygous tofor Zhedong goose site AA, then AB and AA cancould appear in the F2 and F3 generations. However, BB iswas not possible. See Figure 1 for the green annotation. These sites may be candidate SNP sites related to the phenotype of egg- laying.”

Line 123, add the statements of “This process usesused the Sort method ofin Microsoft Excel.”

  1. Sequencing data statistics are not given.

RESPONSE:

Line 429, add the statements of “Table 2. Sequencing data statistics”

  1. PCA must be given for the two breeds.

RESPONSE:

Line 271, add the statements:” Supplementary material:

Supplementary Figure S1. Cross Entropy Line Chart. The minimum value of cross entropy is generally se-lected for population result analysis. There is no obvious population structure in this analysis.

Supplementary Figure S2. Genetic relationship and principal component analysis (PCA). (A) Kinship analysis heat map. The different colors represent kinship. There is no kinship between the samples. In the figure, C repre-sents the original Zhedong goose, B represents F2, and O represents F3.

Supplementary Figure S3. Analyze the Linkage disequilibrium (LD) decal. It is not possible to group using different generations based on the distance in the picture.

Supplementary Figure S4. NJ evolutionary tree. Grouping explicitly based on the distance in the picture is not possible.”

**PCA in three kind geese. In the figure, C repre-sents the original Zhedong goose, B represents F2, and O represents F3**

  1. How were the 3370 candidate SNPs selected?

RESPONSE:

Line 468, the statements of “3370” was changed by “3,370”.

  1. Candidate SNPs with location and p values for association need to be given.

RESPONSE:

Line 432, add the statements of “Table 3. 39 candidate SNPs were screened out in this study.”

Line 178, the statements of “supplementary material table 1” was changed by “table 3”.

  1. The results obtained need to be explained in text and not just mentioned as figures.

Line 162, add the statements of “The above results show that the classical GWAS method cannot achieve the expected effect of the project.”

Line 182, add the statements of “SNP3 and SNP4 have very limited impact on NUP37 protein.”

Line 187, add the statements of “SNP11 will affect the biological activity of NUDT9.”

  1. Introduction and discussion need to be improved greatly.

RESPONSE:

Line 42, add the statements of “DifferentVarious numbers of SNPs have been reported in species of domestic geese have different SNPs reported [3, 4, 5]. The purpose of , and these can possibly be used in live-stock breeding is to obtain better genotypesuperior genotypes.”

Line 297, add the statements of “5.      Gao G, Chen P, Zhou C, Zhao X, Zhang K, Wu R, Zhang C, Wang Y, Xie Y, Wang Q. Genome-wide association study for reproduction-related traits in Chinese domestic goose. Br Poult Sci 2022 63, 754-760.”

Line 214, add the statements of “The egg production of domestic geese is a quantitative trait regulated by multiple genes, which and is affected by environmental factors [22]. The egg-laying characteristic of zhedongthe Zhedong goose is to repeatcomprises repeating the cycle of egg-laying and broodiness. BirdBirds in nature also hashave similar behaviorbehaviors [23]. Broodiness is essential for the breeding of wild birds [24], exceptwith the exception of brood parasit-ism by cuckoos [25]. The artificial domestication of geese is far from the effect ofin hens. The artificial domestication of geese is far from the effect of hens. After years of selection and elimination, hens have completely lost their broodiness ability[12].”

Line 357, add the statements of “22.    Gao G, Gao D, Zhao X, Xu S, Zhang K, Wu R, Yin C, Li J, Xie Y, Hu S, Wang Q. Genome-Wide Association Study-Based Identification of SNPs and Haplotypes Associated With Goose Reproductive Performance and Egg Quality. Front Genet 2021 12, 602583.”

Line 361, add the statements of “23.    Farrar VS, Flores L, Viernes RC, Ornelas Pereira L, Mushtari S, Calisi RM. Prolactin promotes parental responses and alters reproductive axis gene expression, but not courtship behaviors, in both sexes of a biparental bird. Horm Behav 2022 144, 105217.”

Line 365, add the statements of “24.    Norris AR, Martin K, Cockle KL. Weather and nest cavity characteristics influence fecundity in mountain chickadees. PeerJ 2022 10, e14327.”

Line 368, add the statements of “25.    Morelli F, Benedetti Y, Pape Moller A. Diet specialization and brood parasitism in cuckoo species. Ecol Evol 2020 10, 5097-5105.”

  1. Other corrections are marked on the pdf file.

RESPONSE:

I'm very sorry, because I can't use this submission system skillfully. As a result, you cannot see my supplementary file and I cannot see your PDF document. If there has any problem (Whether the content of the paper itself or grammar), please let me know, we will edit again.

  1. English language needs to be improved throughout the manuscript.

RESPONSE:

Thank you for advice. The manuscript was changed many parties and sent to Letpub for language edit again.

Other change

We update the author's email and contributions.

(1)

Jinyan Sun  [email protected]

Shan Yue     [email protected]

Manyu Li    [email protected]

(2)

Guojun Liu and Zhenhua Guo are work out the draft together.

# Zhenhua Guo and Guojun Liu contributed equally to this work and are considered equal first authors.

Reviewer 2 Report

Although the work appears novel there are too many concerns that need to be addressed before the paper can be accepted.

Firstly the title says GWAS but no details of GWAS or the results are presented. GWAS was performed by which software, what model was used?

Sequencing data statistics are not given.

PCA must be given for the two breeds.

How were the 3370 candidate SNPs selected?

Candidate SNPs with location and p values for association need to be given.

The results obtained need to be explained in text and not just mentioned as figures.

Introduction and discussion need to be improved greatly.

Other corrections are marked on the pdf file.

English language needs to be improved throughout the manuscript.

Author Response

(The authors gave the same response as above.)

Round 2

Reviewer 1 Report

The manuscript “Genome-wide association study identifies single-nucleotide polymorphism markers of growth and reproduction traits in Zhedong and Zi crossbred geese” still has some issues that need to be addressed before publishing.

Why is the title “Genome-wide association study identifies …” if in L108-109 you establish that GWAS failed and L162 says that the GWAS method was unable to achieve the project's expected effect? I believe there should be a clarification about this as it is confusing.

L159 what description is incorrect?

L190-196. This is not a discussion

The discussion still doesn’t give possible explanations for each finding, it seems more of a literature review. Link each result with the possible explanation and support this with the available literature.

Figure 2B, 2D, 3B, and 5 need to be improved, the text is too small and is incomprehensible.

Author Response

Revised manuscript 2140918 " Genome-wide association study identifies single-nucleotide polymorphism markers of growth and reproduction traits in Zhedong and Zi crossbred goose"

Dear Editor-in-Chief Pro Ziana Zhang, and Reviewers:

On behalf of my co-authors, we thank you very much for giving us an opportunity to revise our manuscript, we appreciate editor and reviewers very much for their positive and constructive comments and suggestions on our manuscript.

The manuscript has already been corrected and is sent to you for further evaluation.

We would like to express our great appreciation to you and reviewers for comments on our paper. Looking forward to hearing from you.

It's a very worrying time for all of us. I hope you are all in good health, and hoping the Coronavirus pandemic will soon be over.

Thank you and best regards.

Yours sincerely,

Guo

Animal Husbandry Research Institute of Heilongjiang Academy of Agricultural Sciences

368 Xuefu Road, Harbin, P.R.China, 150086

Office Tel: 086-451-87502330

Mobile:  086-13115607125

Responses to Reviewers

Reviewer 1

The manuscript “Genome-wide association study identifies single-nucleotide polymorphism markers of growth and reproduction traits in Zhedong and Zi crossbred geese” still has some issues that need to be addressed before publishing.

  1. Why is the title “Genome-wide association study identifies …” if in L108-109 you establish that GWAS failed and L162 says that the GWAS method was unable to achieve the project's expected effect? I believe there should be a clarification about this as it is confusing.

 RESPONSE:

Thank you very much for your quick and timely response to the revision guidance.

Line 1, the title of “Genome-wide association study identifies single-nucleotide polymorphism markers of growth and reproduction traits in Zhedong and Zi crossbred geese” was changed by “Whole genome resequencing identifies single-nucleotide polymorphism markers of growth and reproduction traits in Zhedong and Zi crossbred geese”.

Line 54, the statements of “a genome-wide association study (GWAS)” was changed by “whole genome resequencing”.

Line 142, the statements of “Genome-Wide Association Study” was changed by “Whole Genome Resequencing”.

Line 6, the statements of “GWAS” was changed by “Whole genome resequencing”.

Line 30, the statements of “GWAS” was changed by “Whole genome resequencing”.

Line 102, the statements of “GWAS” was changed by “genome-wide association study (GWAS)”.

Line 422, the statements of “GWAS” was changed by “whole genome resequencing”.

Figure 1, the statements of “GWAS” was changed by “whole genome resequencing”.

Graphical abstract, the statements of “GWAS” was changed by “whole genome resequencing”.

Attached Figure 1.jpg (378KB, 300DPI)

Attached Graphical abstract.jpg (794KB, 500DPI)

  1. L159 what description is incorrect?

 RESPONSE:

Line 152, the statements of “The description is incorrect” was changed by “The results were disappointing and could not be analyzed according to our grouping.”

  1. L190-196. This is not a discussion

 The discussion still doesn’t give possible explanations for each finding, it seems more of a literature review. Link each result with the possible explanation and support this with the available literature.

 RESPONSE:

Line 183, delete the statements of “Many methods have been explored to reduce the broodiness of geese, including hormone intervention [13, 14], light adjustment [15, 16, 17], genetic improvement [18, 19], and nutri-tion regulation [20].”

Line 183, add the statements of “Many methods have been explored to reduce the broodiness of geese, including hormone intervention [3, 4], light adjustment [5, 6, 7], genetic improvement [8, 9], and nutrition reg-ulation [10].”

Line 193, add the statements of “Our hypothesis is the SNP11 may affect the goose egg-laying trait, and the verification experiment will continue in the future.”

Line 238, add the statements of “Because this change did not cause changes in the advanced structure of the NUP37 protein, and it also does not affect the structures of the binding pockets.”

Figure 2B, 2D, 3B, and 5 need to be improved, the text is too small and is incomprehensible.

RESPONSE:

Attached Figure 2B.jpg (711KB, 300DPI)

Attached Figure 2D.jpg (509KB, 300DPI)

Attached Figure 3B.jpg (748KB, 300DPI)

Attached Figure 5.jpg (1.24MB, 300DPI)

Reviewer 2 Report

The manuscript has improved a lot, however, I still have 2 concerns-

1. The title should be revised as GWAS was not done. Suggested title- Whole genome resequencing identifies single-nucleotide polymorphism markers of growth and reproduction traits in Zhedong and Zi crossbred geese.

2. Line 159. "The description is incorrect". Please modify this sentence. If the description is incorrect it should be corrected.

Author Response

Revised manuscript 2140918 " Genome-wide association study identifies single-nucleotide polymorphism markers of growth and reproduction traits in Zhedong and Zi crossbred goose"

Dear Editor-in-Chief Pro Ziana Zhang, and Reviewers:

On behalf of my co-authors, we thank you very much for giving us an opportunity to revise our manuscript, we appreciate editor and reviewers very much for their positive and constructive comments and suggestions on our manuscript.

The manuscript has already been corrected and is sent to you for further evaluation.

We would like to express our great appreciation to you and reviewers for comments on our paper. Looking forward to hearing from you.

It's a very worrying time for all of us. I hope you are all in good health, and hoping the Coronavirus pandemic will soon be over.

Thank you and best regards.

Yours sincerely,

Guo

Animal Husbandry Research Institute of Heilongjiang Academy of Agricultural Sciences

368 Xuefu Road, Harbin, P.R.China, 150086

Office Tel: 086-451-87502330

Mobile:  086-13115607125

Responses to Reviewers

Reviewer 2

The manuscript has improved a lot, however, I still have 2 concerns-

  1. The title should be revised as GWAS was not done. Suggested title- Whole genome resequencing identifies single-nucleotide polymorphism markers of growth and reproduction traits in Zhedong and Zi crossbred geese.

RESPONSE:

Thank you very much for your quick and timely response to the revision guidance.

Line 1, the title of “Genome-wide association study identifies single-nucleotide polymorphism markers of growth and reproduction traits in Zhedong and Zi crossbred geese” was changed by “Whole genome resequencing identifies single-nucleotide polymorphism markers of growth and reproduction traits in Zhedong and Zi crossbred geese”.

Line 54, the statements of “a genome-wide association study (GWAS)” was changed by “whole genome resequencing”.

Line 142, the statements of “Genome-Wide Association Study” was changed by “Whole Genome Resequencing”.

Line 6, the statements of “GWAS” was changed by “Whole genome resequencing”.

Line 30, the statements of “GWAS” was changed by “Whole genome resequencing”.

Line 102, the statements of “GWAS” was changed by “genome-wide association study (GWAS)”.

Line 422, the statements of “GWAS” was changed by “whole genome resequencing”.

Figure 1, the statements of “GWAS” was changed by “whole genome resequencing”.

Graphical abstract, the statements of “GWAS” was changed by “whole genome resequencing”.

Attached Figure 1.jpg (378KB, 300DPI)

Attached Graphical abstract.jpg (794KB, 500DPI)

  1. Line 159. "The description is incorrect". Please modify this sentence. If the description is incorrect it should be corrected.

RESPONSE:

Line 152, the statements of “The description is incorrect” was changed by “The results were disappointing and could not be analyzed according to our grouping.”

Other change

No
